# Overloaded and Unrestrained: A Qualitative Study with Local Experts Exploring Factors Affecting Child Car Restraint Use in Cape Town, South Africa

**DOI:** 10.3390/ijerph17144974

**Published:** 2020-07-10

**Authors:** Kate Hunter, Amy Bestman, Madeleine Dodd, Megan Prinsloo, Pumla Mtambeka, Sebastian van As, Margaret Mary Peden

**Affiliations:** 1The George Institute for Global Health, The University of New South Wales, Sydney 2042, Australia; abestman@georgeinstitute.org.au (A.B.); mdodd@georgeinstitute.org.au (M.D.); 2Burden of Disease Research Unit, South African Medical Research Council, Tygerberg, Cape Town 8000, South Africa; Megan.Prinsloo@mrc.ac.za; 3Woolworths Childsafe Research and Educational Centre, Cape Town 7701, South Africa; Pumla.Mtambeka@westerncape.gov.za (P.M.); Sebastian.VanAs@uct.ac.za (S.v.A.); 4Department of Paediatric Surgery, University of Cape Town, Cape Town 7701, South Africa; 5The George Institute for Global Health UK, Oxford University, Oxford OX1 2BQ, UK; margie.peden@georgeinstitute.ox.ac.uk; 6School of Public Health and Community Medicine, University of New South Wales, Sydney 2052, Australia

**Keywords:** passenger, restraint, qualitative, child injury

## Abstract

(1) *Background*: Children in South Africa experience significant impacts from road injury due to the high frequency of road crashes and the low uptake of road safety measures (including the use of appropriate child restraints). The current study aimed to assess the feasibility of a child restraint program and to describe factors influencing child restraint use from the perspectives of clinicians, representatives of non-government agencies, and academics in Cape Town, South Africa. (2) *Methods*: Qualitative interviews were conducted with 13 experts from government, academic and clinical backgrounds. Findings were analyzed using the COM-B component of the Behaviour Change Wheel and were grouped by the layers of the social-ecological model (individual, relational, community and societal). (COM-B is a framework to explain behaviour change which has three key components; capability, opportunity and motivation), (3) *Results*: Experts believed that there was a need for a child restraint program that should be staged and multifactorial. Participants described knowledge gaps, perceptions of risk, mixed motivations and limited enforcement of child restraint legislation as key influences of restraint use. (4) *Conclusions*: The results demonstrate potential areas on which to focus interventions to increase child restraint use in Cape Town, South Africa. However, this will require a coordinated and consistent response across stakeholder groups.

## 1. Introduction

In 2004, the World Health Organisation (WHO) published the ‘World report on road traffic injury prevention’ [1] followed by the ‘World report on child injury prevention’ in 2009, which appealed to governments to keep children safe through a reduction in the increasing burden caused by road traffic injuries [2]. These two documents led to the declaration of a Decade of Action for Road Safety (2011–2020) [3] and the inclusion of a Sustainable Development Goal in 2015 to halve the number of global deaths and injuries from road traffic crashes (Target 3.6) [4]. In 2017, WHO Member States developed twelve road safety performance targets—one of which specifically focuses on child road safety and calls for an increase in “the proportion of motor vehicle occupants using safety belts or standard child restraint systems” to close to 100% by the year 2030 [5]. Following the introduction of relevant legislation, the success of such targets relies heavily on the commitment of stakeholders. This includes police, clinicians, and transporters to enforce the law, educate carers, and address any barriers to child restraint use experienced by those responsible for transporting children. 

Despite action at a global level through the WHO, road traffic injuries remain the leading cause of death for young people aged 5–29 years [6]. Across the six WHO global regions, the African region has the highest overall rate of road traffic injuries [6], with rates for children aged 0–4 years at 26.14 per 100,000 population [7]. In South Africa in 2009, the road traffic mortality rates for children aged 0–4 years and 5–14 years were 13.5 and 11.2 deaths per 100,000, respectively [8]. Rates vary across the country. In Cape Town (Metro West), among children aged up to 14 years, the death rate for road traffic crashes is reported at 8.7 per 100,000 population [9]. Of all road traffic deaths in South Africa, approximately 40% are accounted for by pedestrians and 33% by passengers [8]. In the Western Cape (the province in which Cape Town is situated), 63% of households reported taxi (26%), or car, bakkie or truck (37%) as their main modes of travel [10]. The Western Cape Province has the greatest proportion of scholars attending educational institutions by cars, bakkies or trucks compared with South Africa in general (24% versus 12.9%) [10]. 

Child passenger deaths are preventable. The appropriate use of child car restraints significantly reduces child passenger deaths, with a death risk reduction for children aged one to three years of between 74% and 59% [11]. Child restraint use is a core factor of road safety strategies globally and is a listed priority in the South African Road Safety Strategy 2011–2020 [12]. While a law mandating the use of South African Bureau of Standards-approved child restraints for all children under the age of three years exists in South Africa [13], data indicate widespread low rates of use [14,15]. For example, an analysis of specific types of road traffic injuries in Cape Town found that unrestrained passenger injuries were the most common mechanism of road traffic injury for children aged 0–12 months and the second most common for children between 0 and 4 years following pedestrian injuries [16]. Additionally, in 2018, an observational study documenting the use of child restraints among visitors and patients attending the Red Cross Children’s War Memorial Hospital in Cape Town observed only 18% (*n* = 38) of children using some form of restraint, while 33% (*n* = 71) were sitting on the lap of an adult [14]. Multifaceted, education-based child restraint programs have been shown to increase optimal child restraint use (a child correctly secured in a properly installed appropriate child restraint) in Australia [17,18] and the United States [19]. Such multifaceted programs included hands-on education sessions to help parents and carers secure a child in an age- and size-appropriate child restraint and install the restraints in a vehicle, educational resources for early childhood educators to include road safety in their education program, and access to subsidised child restraints and child restraint-fitting services. These programs were shown to increase the use of restraints over and above the introduction of child restraint legislation in Australia [17]. Given the high burden of road-related deaths and serious injury experienced by young children in South Africa, and evidence showing that child passenger safety laws are not adhered to, there is a clear need to better understand factors that impact child restraint use. This includes both the adherence to and enforcement of legislation, and the feasibility of child restraint programs in South Africa. To date, no other published research has explored the feasibility of developing a child travel safety program in South Africa. Using Michie’s Behaviour Change Wheel [20] and a social-ecological approach to child restraint use, this research aimed to investigate stakeholders’ perspectives of behaviours and perceptions related to the ownership and the use of child restraints in Cape Town, South Africa. It was anticipated that results from this qualitative study could inform the development of a child passenger safety program. This study was guided by the following research questions: What factors influence children’s road safety in Cape Town, South Africa?What are the current attitudes and behaviours relating to child restraints in Cape Town and what are the factors that influence this?What type of intervention would have the most potential to increase the use of child restraints in Cape Town?

## 2. Materials and Methods 

Our reporting is guided by the Consolidated Criteria for Reporting Qualitative Research (COREQ) [21].

### 2.1. Recruitment and Data Collection

We identified fourteen experts based in Cape Town, South Africa, comprising police, surgeons, representatives from the Department of Transport, representatives from non-government organisations (NGOs) and academics. A member of our team (M.P.) invited them to take part in the study. Thirteen stakeholders agreed to be interviewed (one was travelling and therefore not available). All the interviews were conducted by M.M.P. in English over a two-week period in June 2018 and took between 45 minutes and one hour. Permission to audio tape the interviews on a Sony digital recorder was obtained and all participants signed consent forms. The semi-structured interview guide was designed by M.M.P. and K.H. based on the objectives of the study (Appendix A). Questions covered the following themes: child road safety in Cape Town; child restraints as an intervention; barriers and facilitators to using restraints; and futuristic interventions for child road safety. Questions were modified slightly based on whether the interviewee was a policy maker (macro), health care worker, practitioner or researcher (meso) or community leader, parent, or caregiver (micro). The interview schedule was piloted prior to commencing the formal interviews, with minor adjustments made primarily to the sequencing of questions. Most questions were open ended. However, probing and follow-up questions were also utilized where appropriate to improve the richness of information. Ethical approval to conduct the study was received from both the University of Cape Town’s Human Research Ethics Committee (HREC 255/2018) and from Oxford University’s Tropical Research Ethics Committee (OxTREC Ref. 534-18).

### 2.2. Data Analysis

All interviews were transcribed using the free version of Express Scribe v 7.03. by M.M.P. Once the quality of transcription was verified by M.D., the 13 transcripts were uploaded into NVivo 12 Pro for analysis. Inductive and deductive thematic analysis was employed by M.D., A.B. and K.H. To describe factors affecting child restraint use, themes were underpinned by both the Behaviour Change Wheel [20] and focused on the COM-B component of the framework [20] and the social-ecological model of behaviour [22]. The COM-B model of behaviour change acknowledges that behaviours are derived from a complex set of interacting factors [20]. It describes three broad factors that influence human behaviour, ‘capability’—an individual’s psychological and physical capacity to engage in the activity concerned; ‘opportunity’—factors that lie outside the individual that make the behaviour possible or prompt it; ‘motivation’—brain processes that energise and direct behaviour [20]. These three factors were then broken down further to highlight important behavioural distinctions that inform the Behaviour Change Wheel [20]; see the first two columns of Table 1 for the full definition for each different category as it relates to car restraint use. The social-ecological model originally developed by Bronfenbrenner [23] has been applied to describe factors influencing injury and to plan injury prevention programs including agriculture safety [24], violence prevention [22], older driver safety [25] and general road safety [26]. The model recognises that behaviour is influenced by the interplay of several factors including those related directly to the individual, their relationships, their community and society. This analysis combines the COM-B and social-ecological models to inform and plan future child road safety programs.

## 3. Results

Participants were from a range of professions including the Western Cape Department of Transport (*n* = 2) (one transport specialist and the other a law enforcement officer), academia (two in the field of public health and one engineering), representatives from child injury non-governmental organisations (*n* = 2) and six clinicians (including paediatric surgeons, physicians, pathologists and nurses). Seven participants were male and six were female. Figure 1 summarises the factors participants described as influencing child restraint use under the constructs of the social-ecological model. Table 1 then presents the six theoretical domains of COM-B Model of Behaviour Change and the enablers and barriers identified by stakeholders for each domain. The following results are presented according to the key themes in each of the domains. 

### 3.1. Individual Factors Influencing Restraint Use 

#### 3.1.1. Perceptions of Risk among Parents and Carers 

The majority of participants described a lack of knowledge around the importance of child restraints as a major reason why parents and caregivers do not restrain their children. Participants stated that although there is some level of understanding of the consequences of a road traffic crash, stakeholders believed that there was a gap in awareness of the increased injury risk severity for an unrestrained child in a crash. Half of the participants said that parents were ‘ignorant’ of the harms for children. 


*“They see that the chance [of a crash] is very remote … The fundamental thing is that people believe for example that they can hold a baby on their lap. People do not understand that at 20km/hr if you come to a sudden halt that the child is going to go through the windscreen or at the very least be very severely injured. People just do not—there is no understanding of the risks.”*
Participant_1 (Government)

For parents who owned child restraints, experts believed that many chose not to use them or used them incorrectly due to knowledge gaps. These included not having a restraint appropriate for the child’s height and weight or perceptions of people’s misunderstandings associated with the harm caused by seat belts/child restraints, rather than the injury averted:
“It’s a total myth. You can get nasty bruising of course, but if you get nasty bruising that is a sign that if you hadn’t been restrained you would have probably hit something with a lot more force and had a much more serious injury…. So people don’t really think things through. And South Africans in particular have an extraordinary ability to make excuses for their behaviour in traffic.”Participant_12 (Road safety engineer)

#### 3.1.2. Perceptions that Convenience Overrode Safety

Participants also had perceptions that the non-use of restraints was a choice that prioritised convenience over safety:
“Then there is an attitude of convenience. It is far more convenient to have a child bouncing around on your knees on the front passenger seat than having to turn around and deal with someone who is restrained behind you. So I think a lot of it is inconvenience versus convenience and laziness versus diligence.”Participant_11 (Paediatric surgeon)

#### 3.1.3. Perceptions of Risk among Older Children 

Stakeholders also reported perceptions of a lack of knowledge among children not understanding the risks of not using child restraints and suggested the need to include children in road safety education.


*“Children need to understand why they use seat belts and car seats. Then they will be more willing [to use] them”*
Participant_4 (NGO for road safety and children)

Some participants described children not wanting to wear seatbelts and arguing with parents or placing seatbelts “*around their backs*” to avoid wearing them. 

#### 3.1.4. Perceptions of Financial Barriers 

Participants also perceived a possible financial barrier to car restraint usage. However, while some participants stated that the financial cost of child restraints could impact access to restraints, others stated there was no excuse (that if a family could afford a vehicle, they should prioritise child car restraints). 


*“What I have heard is that our people are poor and poor people can’t afford expensive car seats and car seats are expensive. So are the people poor? Yes, they are poor. Are they rich enough to afford a motor car to drive in? Yes, they are. Can they put a tank of petrol in that car? Now you know it is R700 for a tank of petrol, which actually is the price of a cheap car seat. … don’t accept poverty as an excuse. If you are going to take the responsibility of driving a car with a child in it I really think that you need to supply that. So poor people, but rich enough to afford a car, if you are rich enough to afford a car, you can afford a car seat.”*
Participant_11 (Paediatric surgeon)


*“There is the poverty aspect of whether they can afford a car seat or not but I think that is being made unnecessarily complicated because there are ways to get cheap car seats or free car seats. Car seat banks and things like that.”*
Participant_3 (Emergency medical officer)

### 3.2. Relationship

Participants described drivers “*giving in*” to children who did not want to be restrained or, in some instances, larger families forgoing car seats to carry all passengers in the car. 

### 3.3. Community 

#### 3.3.1. Public Campaigns

Participants described a lack of public campaigns and emphasis around the crash consequences and impacts for children with no restraints. Some stated that the focus to date of public discussions was around child pedestrian injury rather than passenger injury, which impacted on public perceptions and therefore did not reinforce the use of car restraints. This was also linked to a lack of publicly available information around correct child restraint use. 

#### 3.3.2. Modes of Transportation Impact Capacity to Use Restraints and Risk Perceptions 

All participants noted that the common modes of transport in South Africa acted as key barriers to the use of child restraints. Different forms of transportation included ‘bakkie’ (a small truck with an open body and low sides), ‘scholar transporters’, ‘private minibus taxis’ and ‘private vehicles’. Participants stated that some government policies encouraged the use of transport with not enough seats for the number of children. This form of transportation—usually a minibus and locally referred to as ‘scholar transport’—is government subsidised for children who walk more than 5 km to school or paid for by parents who choose to send their children to schools away from their local area. Consequently, these scholar transporters usually transport at least double the number of children than there are seats in the vehicle. 


*“I’d say there are three different ways [children are transported]: There is in a private car—so my son has gone to Cape Town today with a school trip and we had to specify to the teachers which parents are allowed to take him because the majority of parents in his class do not restrain their kids and they will have a sedan with seating for five and they will have six or seven kids in there just on the school run regularly. So private cars there is a minority of children who are transported properly restrained both anecdotally and from a study I did a couple of years ago where we stood on the side of the road and we had observers—about 60% of adults had seat belts but only 20% of kids … my anecdotal experience in private cars it is either unrestrained or the majority are just with a seat belt not with proper fitted car seat. The second is bakkies—so especially in rural areas—bakkies full of kids either with a cover on the back or without—15 on the back of a bakkie is not unusual. And third is the minibus taxi either in Cape Town minibuses with mixed passenger loads or the dedicated school buses and the same out in rural areas. And they are usually overloaded and unrestrained.”*
Partcipant_3 (Emergency medical officer)

It was noted that due to the high rate of pedestrian-related motor vehicle injuries, parents and carers often considered it safer for children to be transported unrestrained on the back of a ‘bakkie’ rather than allowing them to walk on the streets.


*“There is no consistent focus to police it as such. And the realities—you have got to ask is are you going to rather allow those children to walk 10–12 km to school every day or are they going to transport them on the bakkie. You know if you look at the number of pedestrian kids killed then actually it is almost safer … transporting them on a bakkie than allowing them to walk.”*
Participant_ 7 (Traffic officer)

#### 3.3.3. Perceptions of Socially Embedded Practices

Consistent with the structural factors described above, there was a community acceptance of issues including overcrowding in cars, or not using car restraints. Additionally, participants described the acceptance of risk taking or harmful behaviours that increase the risk of a crash:
“People have a profile of taking risk thinking that it will not happen to them and thinking that their children are strong enough. Not understanding the vulnerability specifically of children with underdeveloped or less developed spines and skulls as adults. And it is that good old stubbornness of people in countries that have a history of frontier people and that is just that they won’t be told what to do.”Participant_ 4 (NGO for road safety and children)

Just under half of the participants described the acceptance of adults restraining children on their laps. Those working in clinical settings described the poor outcomes of children involved in crashes who were restrained in this manner. However, two participants noted that this learnt behaviour was often ingrained into practice for many families. For example, one participant described the multigenerational legacy of this behaviour:
“My parents had me on their laps, and their parents had them on their laps, so why shouldn’t I have my kid on my lap in the front seat.”Participant_13 (Forensic pathologist)

Three participants also expressed a perception that the issue was less about not knowing the importance of and how to use restraints and more an issue of community attitudes. For example, one participant said *“the main problem is not so much ignorance as apathy”.*

### 3.4. Society

#### Legislation and Enforcement 

Participants described the need for increased policy and enforcement around child restraint use, with one participant stating, “*you need policy and laws to underpin good practice*”. 


*“No one has ever told them, there is no public education about it, there is a law but it’s not really well enforced and if it is enforced they don’t really know understand [sic] why they have to do it.”*
Participant_3 (Emergency medical officer)


*“There should be consequences to the drivers who drive the vehicles because there is clearly—we need to bring public transport into the party—currently they are excluded. Taxis—it is not legal for them to put car seats in. We need to change that. And the practicality of that I fully understand, but we need to bring the laws in line with what is acceptable levels of safety for children and then we need the enforcement. There needs to be consequences for the driver”*
Participant_4 (NGO for road safety and children)

### 3.5. Feasibility of a Child Restraint Program

Experts believed that an intervention for child restraints would be of value and that child passenger safety should be made a priority (following pedestrian safety). Experts said that interventions should be multisectoral and engage existing resources such as nurses who have contact with families, schools, government and NGOs. Potential interventions included education and awareness raising of the public, parents, children and those working in the health sector, car seat banks, and regulation.


*“… enforcement plays a very important role, but they are currently over-stretched. So there needs to be a multisectoral response to the issue from government, corporates such as health insurance firms, drivers of minibus taxis and scholar transporters as well as parents and children.”*
Participant_2 (Child injury researcher)


*“I think the biggest one is the lack of knowledge. People just do not understand what seat belts do and what child restraints do. I teach our second years transport course and one of things [sic] I explain to them is the three impacts in a crash. And they have never heard of this before. And every year I have students coming to me to say I didn’t know that and so now I will never get into a vehicle without a seat belt again. So if we can bring that education earlier on in their lives and if we can extend that education to parents, I think that would make a difference. I think that parents even such highly qualified parents driving around Stellenbosch whose children are unrestrained if they could just understand what happens in the event of a crash, I think that that would be a big win.”*
Participant_12 (Road safety engineer)


*“There is enough money in private and government schools to have one child in a seat in a minibus. I think that if you can afford a car—every car has seat belts—you can seat the bigger child in the seat belts and if you can afford a car you can afford a car seat. There are also charitable donations of car seats and booster seats—for people who have a 20-year-old beaten up car—there are ways to get them.”*
Participant_3 (Emergency medical officer)

Participants suggested a staged approach to a targeted child car restraint program. This could involve programs that first focus on children transported in private vehicles, then scholar transporters and then programs targeting bakkies. 


*“I think the middle bits harder where there are bakkie loads of kids who if they don’t get into the bakkie in a rural area they don’t get to school. And I don’t have a solution to that. I think that it is a gradual societal change and educating people.”*
Participant_3 (Emergency medical officer)

## 4. Discussion

To our knowledge, this is the first study that reports on semi-structured interviews with stakeholders (clinicians, Department of Transport representatives, representatives from non-government organisations and academia) regarding their perspectives of factors influencing child restraint use in any African country. While road safety is recognised as a key cause of injury among children in Africa, no study has previously sought similar input. Our results suggest that stakeholders believe that there is a critical road safety issue regarding a lack of appropriately restrained children and the consequent injury risk. Stakeholders believed that there is a role for the development of a dedicated child restraint program in South Africa. Acknowledging the complex nature of how children travel in Cape Town using multiple modes of transport (private cars, scholar transporters or minibuses and ‘bakkies’), stakeholders recommended developing a staged approach where the first programs target children travelling in private vehicles before working with scholar transporters (minibuses) and then ‘bakkies’.

There are both similarities and points of difference between our findings and those reported in previous qualitative studies [27,28,29,30]. Working through the COM-B model, in keeping with other studies [31,32], the current study highlighted stakeholder beliefs that knowledge and risk perceptions are key factors influencing an individual’s capacity to appropriately use child restraints. Further, we reported stakeholders’ description of perceptions of motivation such as believing that convenience is being placed over safety practices (restraint use), which supports studies reported by Bhaumik et al., 2020 [30]. While previous qualitative studies focused on parents’ and carers’ knowledge and attitudes [30], our study intentionally sought perceptions of other stakeholders; those who would be responsible for developing, delivering and/or evaluating a child restraint program. Our application of combining the social-ecological model and the COM-B model of behaviour is a novel approach to better understanding factors influencing restraint use. We note the COM-B model has been used previously [33] to explain restraint use among high and low socioeconomic groups and among culturally and linguistically diverse participants in Australia. However, we bring additional layers, considering individual, relationship, community and societal factors that could also influence restraint use (Figure 1). The value of considering each of these layers when developing a child restraint program is that it facilitates the identification of potential programs and partners to ensure that community-based programs address perceptions of community norms and influences. Further, by applying the COM-B model, we can identify types of interventions most likely to be effective, using literature around different behaviours [34]. This approach, therefore, can inform the types of interventions most likely to affect child restraint use in this South African context. The COM-B model describes behaviour as the result of the “interacting systems” [34] (p. 11) of capability, opportunity and motivation [34]. Michie et al. state that behaviour change will involve changing one or more of these elements “in such a way as to put the system into new configuration and minimise the risk of reverting” [34] (p. 11). We propose a range of targeted behaviours that act as a starting point to address the barriers to child restraint use (Table 2). These have been presented as part of a matrix with the COM-B elements and the themes identified in this research. The final column in Table 2 contains the intervention functions that relate to the components of the COM-B model specified by Michie et al [20]. It is important to ensure that the implementation of any program is carried out within the context of the local setting in order to ensure that work does not exacerbate inequity. We recommend that interventions are developed and selected based on affordability, practicality, effectiveness (including cost-effectiveness), acceptability, minimal unwanted side effects or consequences and equity (known as the APEASE criteria developed by Michie et al. [34]. 

The next stage in this research will be to develop a suite of strategies that considers the intervention functions and aims to address the factors restricting restraint use identified in our study. This relates to capability (not knowing the benefits of restraints and conversely the injury risk of non-restraint use and not knowing when and how to use restraints properly); opportunity (mixed availability of child restraints and mixed perceptions of impact of cost as a potential barrier; and access, or a lack of access, to vehicles such as private cars, taxis versus scholar transporters and bakkies that have varying capacities to accommodate the use of restraints); and motivation (there was an identified need for a safe travel program and mixed perceptions of parents’, carers’ and those responsible for transporting children’s motivations for using child restraints). While our narrative synthesis of factors affecting restraint use identified several universal factors [30], this study has highlighted the importance of considering context. Of note, this study highlights the importance of context when both examining factors influencing child restraint use and considering the feasibility of a child restraint program. Participants described varying factors such as the type of vehicle and travel options available to families, and their perceptions of community norms (overcrowding vehicles, a lack of enforcement of restraint legislation, and choosing convenience over restraint use) that all contribute to restraint use in Cape Town, South Africa. Further, while the value of legislation and, importantly, enforcement of legislation has been associated with increased restraint use [35], this study reported limited enforcement to date in Cape Town. This research reported that child car restraint use is both complex and multifactorial, requiring engagement with a range of stakeholders and calls for a coordinated approach across stakeholder groups to increase car restraint use for children in Cape Town, South Africa. Finally, while this research specifically examined child restraint use in Cape Town, it should not be ignored that the use of child restraints does not reduce the number of crashes that occur. Given the burden of road traffic injury for children in South Africa [7], there is a clear need for crash prevention. Interventions promoting those of child restraints should also seek to address the broader factors that contribute to increased injury risk. For example, while an intervention may address risk perceptions around the use of child restraints, this could extend to the benefits of adult restraint use and increase overall knowledge of factors that contribute to crashes. This study has several limitations. First, this preliminary work contained a small number of participants. As such, this study, therefore, is not to be interpreted as representative, and serves as the first stage in developing a safe travel program. We note that six participants were paediatric clinicians. Their insights provide perspectives of factors that they perceive have influenced restraint use and associated injury risk. Further, while highlighting the importance of context, this study is specific to Cape Town, South Africa, so the issues raised may not be transferable to other, possibly rural, settings even within South Africa. Finally, this study highlighted key factors that stakeholders perceived as motivators to use restraints, and the next steps to inform the development of child restraint program will be to explore factors influencing restraint use from the perspectives of parents and carers and those responsible for transporting children. Once obtained, that information can be triangulated with these data to better inform any future child road safety programs in Cape Town. 

## 5. Conclusions

This research has described stakeholders’ perceptions of the barriers and facilitators that currently affect child restraint use in Cape Town, South Africa (the behaviour). We propose potential interventions to improve the behaviour by addressing key barriers, including a lack of perceived risk associated with not using child restraints, relational factors such as the child not wanting to be fully restrained and socially embedded practices that prevent the use of restraints such as overcrowding of vehicles and a lack of enforcement of child restraint legislation. The research, conducted with local experts, reported that stakeholders see a need for a child restraint program and that a program should be staged, multifactorial and targeted across each of the domains of the COM-B and within each layer of the social-ecological model. Finally, this paper highlights the importance of context to inform the development of a road safety program. 

## Figures and Tables

**Figure 1 ijerph-17-04974-f001:**
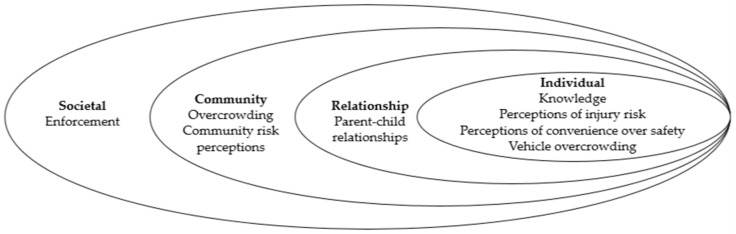
Social-ecological model applied to describe factors affecting child car restraint use from participant interviews.

**Table 1 ijerph-17-04974-t001:** Descriptors of child restraint use in the COM-B domains and themes developed (enablers and barriers) from participant interviews in Cape Town.

COM-B ^1^ Domains	Descriptors	Enablers	Barriers
**Capability**	**An individual’s knowledge and skills to use child restraints**		
Physical	Capacity to use child restraints.	Ablity to physically use child restraints.	Unaffordability of child seats; The incorrect use of child restraints; Lack of seatbelts in vehicles; Cars that are not roadworthy; Overcrowding in cars, minibuses (scholar transporters) or bakkie ^2^ transport.
Psychological	Capacity to engage in necessary thought processes around child restraint use.	Education from health care workers in antenatal phase; Education of school children; Parental education.	Lack of awareness/education on how to restrain children; Acceptance of restraining children on an adult’s lap (multigenerational behaviour); Misunderstanding on the harms of restraints (i.e., bruising/cuts in crashes); Lack of awareness of increased injury risk from not using restraints.
**Opportunity**	**The external factors that enable or act as barriers to the use of child restraints**		
Physical	Environmental factors that prompt or promote the use of child restraints (outside the individual).	Provision of car seats (free or cheap).	Lack of legislation enforcement; Lack of legislation for children over three years; Lack of information on correct child restraint use.
Social	Factors (including cultural) that shape the way individuals conceptualise the use of child restraints and the degree to which these impact on the behaviour.	Public education around restraint use.	Lack of public awareness of consequences/impacts of no restraints in crash; Focus to date has been on child pedestrian injury not passenger injury.
**Motivation**	**Internal processes that direct behaviour relating to the use of child restraints**		
Automatic	Unconscious brain processes (i.e., habit and emotion) that energise and direct behaviour relating to child restraints. Can be increased through habit forming or imitative learning.	Demonstration of appropriate restraint use to parents and children.	Risk taking profiles/thinking; Social acceptance of overcrowding in vehicles; Acceptance of no restraints.
Reflective	Conscious brain processes (i.e., goals, decision making) that energise and direct behaviour relating to child restraints. Can be achieved through increased knowledge and understanding of the behaviour.	Embedding training into nurse’s role; Education for parents and children.	Driving behaviours that lead to crashes (i.e., speeding)—more severe crashes if no restraint is used; Perception that children are at higher risk as a pedestrian rather than as a passenger in overcrowded vehicles and/or being unrestrained.

^1^ = COM-B includes an individual’s capability, opportunity and motivation to describe factors influencing behaviour. ^2^ = a colloquial term in South Africa for a light truck or pickup truck.

**Table 2 ijerph-17-04974-t002:** The targeted behaviour participants described to increase child restraint use in Cape Town and the intervention types identified as most likely to be impactful by the Behaviour Change Wheel [34].

COM-B Domain	Targeted Behaviour	Intervention Types
**Capability** (Psychological)	**Individual factors** Understanding the risks of not using seatbelts/restraints Knowledge on how to use seatbelts and/or restraints. Knowledge of when to transition from one restraint to the next	Education Training Enablement
**Opportunity** (Physical)	**Individual factors** Access to a child car restraint **Relationship/community factors** Access to safe transport options Reduced overloading	Restriction Environmental restructuring Enablement
**Opportunity** (Social)	**Individual factors** Attitudes around staying seated while travelling **Relationship/community factors** Increasing acceptance that seatbelt and restraint use are social norms **Societal factors** Enforcement of legislationAccess to safe transport options (with car seats)	Restriction Environmental restructuring Enablement
**Motivation** (Automatic)	**Individual factors** Automatic reaching for seatbelt Automatic restraining a child in a car seat	Persuasion Incentivisation Coercion Environmental restructuring Modelling Enablement
**Motivation** (Reflective)	**Individual factors** The use of seatbelts The use of restraints Believing there will be consequences **Societal factors** Perception of enforcement of restraint rules	Education Persuasion Incentivisation Coercion

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
