# Peer review of "Overloaded and Unrestrained: A Qualitative Study with Local Experts Exploring Factors Affecting Child Car Restraint Use in Cape Town, South Africa"

_ijerph, 2020, doi:10.3390/ijerph17144974_

Round 1

Reviewer 1 Report

Interesting and valuable research by the authors. I understand that a lot of networking, time and resources are required to do such exploratory research. I really enjoyed the read. 

Line 24 - I would suggest you include the name 'the COM-B component of the Behaviour Change Wheel' in the Abstract inform the reader of the measure used.

Line 42 - Not sure we need to know Target 8.  Also be mindful of reporting numbers consistently throughout the manuscript (8 or eight).

Line 129 - 6 were clinicians out of 13 participants (46% is nearly half) perhaps consider how this may impact on the findings in terms of any bias (e.g factors identified by them as important...or even some that may not have been identified) and consider including some brief information on their exposure to child occupant travel (maybe in participants section).  As a reader it would add value to know that these opinions stem from adequate exposure to parents and child occupant travel practices.

Figure 1.  As it comes up prior to Table 1 that has more information I would suggest you add 'Vehicle' overcrowding and 'Injury' risk to assist the reader.

Table 1.  I may have missed it but shouldn't it include 'lack of awareness of increased injury risk from not using a CRS'? perhaps under Psychological capability.  Another suggestion would be to add 'the higher pedestrian risk than overcrowding in vehicles (including Bakki) in the motivation section as part of the social acceptance note as it comes up in the text and is appears to be an important contextual factor that contributes to the social acceptance?

Line 149  - I struggle with the flow from the previous paragraph 'stakeholders believed there was a gap in the awareness of increased injury risk...' line 146 to this statement 'the fact that the consequences are severe' Doesn't this contradict the earlier paragraph?  May need a sentence prior to the statement to guide the reader to this citation.

Line 181 - the extraction of statements could exclude comments like 'So you know' don't accept poverty as it slows the reader flow.

Line 218 - spacing error 

Line 237 - some participants?? can you be more specific on the strength of these statements (also line 240)

Line 251 - Please reword and check grammar for this paragraph from 'They just don't get the risk, clearly... what? so that interview content is easier to follow. 

Great discussion and support the recommendations of a staged-approach to increase CRS use.

Line 310 - I would be careful using the word 'convenience' in the context of this study unless it is specific in the measures used.  It tends to suggest easier choices are being made which may not be the case for the parents.

I commend authors researching opportunities for a staged, multi-factorial approach and for including next steps in the discussion that include parents input.  It acknowledges the limitations of the perspectives of professionals as third parties and also gives the reader confidence in the efficacy of a tailored approach by including the 'customer' in the design.  Thank you for the read. I look forward to following the next stage of the research.

Reviewer 2 Report

Thank you for the opportunity to review the manuscript titled “Overloaded and unrestrained: A qualitative study exploring feasibility of a child car restraint programme in Cape Town, South Africa”. This manuscript examines the multi-level factors related to age-appropriate restrained system use in South Africa and the potential intervention.  Overall, based on the conceptual framework, this manuscript is written with clear study questions, results and discussion. However, some big concerns in the methodology that need to be addressed. My review of the manuscript and recommendations are enclosed. 

Introduction

The authors did a reasonably good job in describing the epidemiological data for the transportation related children injury in Africa and South Africa. However, what’s the main transportation modes among children in Cape Town? Private car or minibuses? The clarification on this question may help audience to understand the significance of the research questions/topic. Because if walking is the main mode for children use for going to school or traveling, the future intervention should focus on the pediatric pedestrian injury prevention. Additionally, presenting the % of children used each mode of motorized vehicle for traveling also can help audience understand if the key stakeholders in the method part are appropriate to be included in this study.

The three research questions more focused on examining the risk factors of not appropriate children restrain use and developing a future children injury intervention program based on those identified risk factors, rather than exploring the feasibility of an intervention program. As a feasibility study of an intervention program usually addresses the acceptability, implementation, adaptation, expansion, or integration of an intervention program. I did not notice the results or the participants’ responses addressed feasibility. Thus, I suggest the authors may change the research focus or the title.   

Methods

The biggest concerns is whether the key stakeholders for the targeted research topic have been identified and included in this study.

  1. The authors described they identified 14 experts at the beginning and 13 of them agreed to participate the study, however, no further details about these 13 experts were stated. We even don’t know if these experts’ work scope target on children transportation safety and if their job duties are related to the children injury. Without the information about their experts, we don’t know if the responses were from their personal experience or from their working experience they witnessed.

  1. The key stakeholders, including parents/caregivers and/or mini-bus or other public transportation drivers, community leaders or school educators were not included in the study. Although the authors mentioned in their limitations that these experts were the motivators who should be included for this first study. The future study would include the caregivers, the current experts did not the ideal respondents for the research questions and objectives that authors stated in the background section. No matter whether this study focused on examining the feasibility of a children injury program or identifying risk factors of not using restrain system, the parents and drivers from the public transportation the children often used in Cape Town should be very important key stakeholders. Their perceptions play a critical role in reporting the “real” individual risk factors or barriers that impeded the parents/caregivers from using the children restrain system and what type of future intervention meets their expectation for the families and communities.

Thus, I strongly suggested the authors add more key stakeholders (e.g., parents) to the study for this research topic or clarify this study is exploring the perceptions on risk factors from the children injury prevention experts at the beginning (e.g., title, abstract, introduction).

The citation from the respondents in the text are not in a standard format. For example, some citations are using italic, but some citations are not. The citations from respondents should differentiate from the normal texts in each paragraph so that the audience can know which evidences are from respondents, which ones from authors’ summary.  Plus, some risk factors (e.g., 3.1.3) lack of citations from respondents as evidence.

Table 1 COM-B model did not provide the key component of “opportunity”, why? Not applicable?

Round 2

Reviewer 2 Report

I read the authors' responses and modified version of the manuscript. Since the author clarified their focus with the updated title, and new yellow highlighted content addressed my concerns. I don't have further questions regarding this manuscript at this moment. I recommend this manuscript to be accepted.